# Automatically Identifying and Interpreting Sparse Circuits with Hierarchical Tracing

## Abstract

We present a novel approach to Transformer circuit analysis using Sparse Autoencoders (SAEs) and Transcoders. SAEs allow fine-grained feature extraction from model activations, while Transcoders handle non-linear MLP outputs for deterministic circuit tracing. Our Hierarchical Tracing method isolates interpretable circuits at both local and global levels, enabling deeper insights into tasks like subject-verb agreement and indirect object identification. Additionally, we introduce an automated workflow leveraging GPT-4o for scalable circuit analysis. This framework provides a clearer understanding of Transformer model behavior and its underlying mechanisms.

## 1 Introduction

Recent years have seen the rapid progress of mechanistically reverse engineering Transformer language models (Vaswani et al., 2017). Conventionally, researchers seek to find out how neural networks organize information in its hidden activation space (Olah et al., 2020a; Gurnee et al., 2023; Zou et al., 2023) (i.e. features) and how learnable weight matrices connect and (de)activate them (Olsson et al., 2022; Wang et al., 2023; Conmy et al., 2023) (i.e. circuits). One fundamental problem of studying attention heads and MLP neurons as interpretability primitives is their polysemanticity, which under the assumption of linear representation hypothesis is mostly due to superposition (Elhage et al., 2022; Larson, 2023; Greenspan & Wynroe, 2023). Thus, there is no guarantee of explaining how these components impact model behavior out of the interested distribution. Additionally, circuit analysis based on attention heads is coarse-grained because it lacks effective methods to explain the intermediate activations.

Probing (Alain & Bengio, 2017) in the activation for a more fine-grained and monosemantic unit has succeeded in discovering directions indicating a wide range of abstract concepts like truthfulness (Li et al., 2023) and refusal of AI assistants (Zou et al., 2023; Arditi et al., 2024). However, this supervised setting may not capture features we did not expect to present.

Sparse Autoencoders (SAEs) (Bricken et al., 2023; Cunningham et al., 2023) provide a promising alternative for unsupervised feature extraction from superposition. They offer a new perspective on understanding model internals by interpreting the activation of SAE-derived features. This raises an important question: **How can we effectively leverage SAEs for circuit analysis in Transformer models?** To address this, we introduce several innovations in this area. Compared to previous work (Cunningham et al., 2023; He et al., 2024; Marks et al., 2024), our main contributions are as follows:

- We propose a novel framework that utilizes **Transcoders**, generalized forms of SAEs, to overcome the non-linearity of MLPs in Transformer models. Transcoders allow for sparse decomposition of MLP outputs, enabling fine-grained circuit analysis while maintaining deterministic connections between upstream and downstream features.

- We introduce a fully automated **Hierarchical Tracing** methodology to streamline the discovery and interpretation of circuits at both local and global levels, by tracing the flow of information based on sparse features extracted by SAEs and Transcoders.

- We demonstrate the effectiveness of our approach by applying it to tasks including subject-verb agreement and indirect object identification, offering more detailed insight into how each single SAE feature contributes to a desired behavior.

## 2 EXTRACT SPARSE FEATURES WITH SAEs AND TRANSCODERS

### 2.1 SPARSE AUTOENCODER FEATURES AS ANALYTIC PRIMITIVES

Sparse Autoencoder (SAE) is a recently emerging method to take features of model activation out of superposition (Elhage et al., 2022). Existing work has suggested empirical success in the interpretability of SAE features concerning both human evaluation (Bricken et al., 2023) and automatic evaluation (Bills et al., 2023b).

Concretely, an SAE and its optimization objective can be formalized as follows:

$$
\begin{aligned}
f &= \text{ReLU}(W_E x + b_E) \\
\hat{x} &= W_D f \\
\mathcal{L} &= \|x - \hat{x}\|_2^2 + \lambda \|f\|_1,
\end{aligned}
\tag{1}
$$

where $W_E \in \mathbb{R}^{d_{\text{SAE}} \times d_{\text{model}}}$ is the SAE encoder weight, $b_E \in \mathbb{R}^{d_{\text{SAE}}}$ encoder bias, $W_D \in \mathbb{R}^{d_{\text{model}} \times d_{\text{SAE}}}$ decoder weight, $x \in \mathbb{R}^{d_{\text{model}}}$ input activation. $\lambda$ is the coefficient of L1 loss for the balance between sparsity and reconstruction. We refer the reader to Appendix B for implementation details.

We train Sparse Autoencoders on GPT-2 (Radford et al., 2019) to decompose *all modules that write into the residual stream* (i.e. Word Embedding, attention output and MLP output), allowing us to compute cross-layer contribution.

### 2.2 ADDRESSING MLP NON-LINEARITY WITH TRANSCODERS

The dense and non-linear nature of MLPs in Transformers complicates the sparse attribution of MLP features. Observing clear, informative mappings between MLP neurons and learned SAE features is often challenging due to this non-linearity, which disrupts connections between upstream SAE features and MLP outputs.

To mitigate this issue, we introduce Transcoders (proposed by Dunefsky et al. (2024) as contemporary work)—generalized SAEs that decouple the input and output, enabling predictions of future activations based on earlier model states. Transcoders take pre-MLP activations and generate a sparse decomposition of MLP outputs. The optimization objective for a Transcoder is expressed as follows:

$$
\begin{aligned}
f &= \text{ReLU}(W_E x + b_E) \\
\hat{y} &= W_D f \\
\mathcal{L} &= \|y - \hat{y}\|_2^2 + \lambda \|f\|_1,
\end{aligned}
\tag{2}
$$

This differs from the SAE formulation (Equation 1) primarily in that the label activation $y \in \mathbb{R}^{d_{\text{model}}}$ is independent of the input activation $x$.

By employing Transcoders, the generation of MLP output features (termed Transcoder features) becomes **deterministic**. When assessing how an upstream feature $f_i^{\mathcal{S}}$ contributes to a downstream feature $f_j^{\mathcal{T}}$ of Transcoder $\mathcal{T}$, the relationship holds as $f_j^{\mathcal{T}} = f_i^{\mathcal{S}} \left( W_E^{\mathcal{T}} W_D^{\mathcal{S}} \right)_{ji}$. The term $\left( W_E^{\mathcal{T}} W_D^{\mathcal{S}} \right)_{ji}$ remains constant across different inputs, establishing **edge invariance** between upstream and downstream features.

This means that if a primary upstream contributor activates under a different input, we can reasonably expect the corresponding downstream feature to activate as well, unless countered by new resistances (i.e., upstream features with negative contributions).

In contrast, MLPs lack such invariant connections, as any linkage from upstream to MLP outputs is ambiguous. Consequently, we can only apply linear approximations to capture these connections under localized changes.

## 3 ISOLATING INTERPRETABLE CIRCUITS WITH HIERARCHICAL TRACING

We have extracted sparse representations of model activations using Sparse Autoencoders (SAEs) and Transcoders. This section introduces a novel method called *Hierarchical Tracing*, which isolates and evaluates a connected computational subgraph of key SAE / Transcoder features related to any output of interest in a scalable and generalized manner. The goal is to trace interpretable circuits that provide insights into the role of these features in the model's predictions or behavior.

### 3.1 FORMULATION

**Forward Pass as a Computational Graph.** The forward pass of a neural network $M$ can be formalized as a computational graph $\mathcal{G} = (\mathcal{V}, \mathcal{E})$, representing the flow of computation by organizing operations and variables into a directed acyclic graph (DAG), as described by Owhadi (2022). Each node $v \in \mathcal{V}$ corresponds to a model activation $a_v$, which exists in an activation space $\mathcal{A}_v$. Each directed edge $e = v \rightarrow u \in \mathcal{E} \subset \mathcal{V} \times \mathcal{V}$ encodes the functional dependence of $u$ on $v$ via a mapping $g_e$.

For any non-leaf node $u \in \mathcal{V}$, the activation $a_u$ is determined by the activations of its predecessor nodes $v$, according to:

$$a_u = \otimes_{v \rightarrow u} g_{v \rightarrow u}(a_v), \tag{3}$$

where $\otimes$ represents the aggregation of inputs from all incoming edges to node $u$. This formulation captures the structured flow of information through the network during the forward pass and sets the foundation for a deeper analysis of node interactions.

**Path-based Gradient Computation.** We adopt a path-based approach to gradient computation, which decomposes the gradient into contributions from individual paths in the computational graph. Consider a single path $P$ connecting two nodes $v \in \mathcal{V}$ and $u \in \mathcal{V}$. The gradient of activation $a_u$ with respect to $a_v$ along this path $P$ is given by:

$$\begin{aligned}
\nabla_v a_u \Big|_P &= \nabla_{u_n} a_u \cdot \nabla_{u_{n-1}} a_{u_n} \cdot \ldots \cdot \nabla_v a_{u_1}, \\
&= \prod_{e \in P} \nabla g_e,
\end{aligned} \tag{4}$$

where the product of gradients is taken over all edges $e$ along path $P$. This expression captures the contribution of a specific path to the total gradient.

The total gradient of $a_u$ with respect to $a_v$ is then the sum of gradients across all possible paths between $v$ and $u$:

$$\nabla_v a_u = \sum_P \nabla_v a_u \Big|_P.$$

This path-based decomposition enables us to attribute the influence of individual paths within the graph, providing a more granular view of how specific subgraphs contribute to the output.

### 3.2 HIERARCHICAL TRACING

**Mounting SAEs and Transcoders.** The sparse features extracted by SAEs and Transcoders are initially absent from the computational graph formed by the original model forward pass. To assess the causal effect of these features, we introduce the concept of *mounting* them into the computational graph, which embeds the encoding and decoding processes of SAEs and Transcoders within the graph, allowing us to trace the flow of information through these components to make features involved.

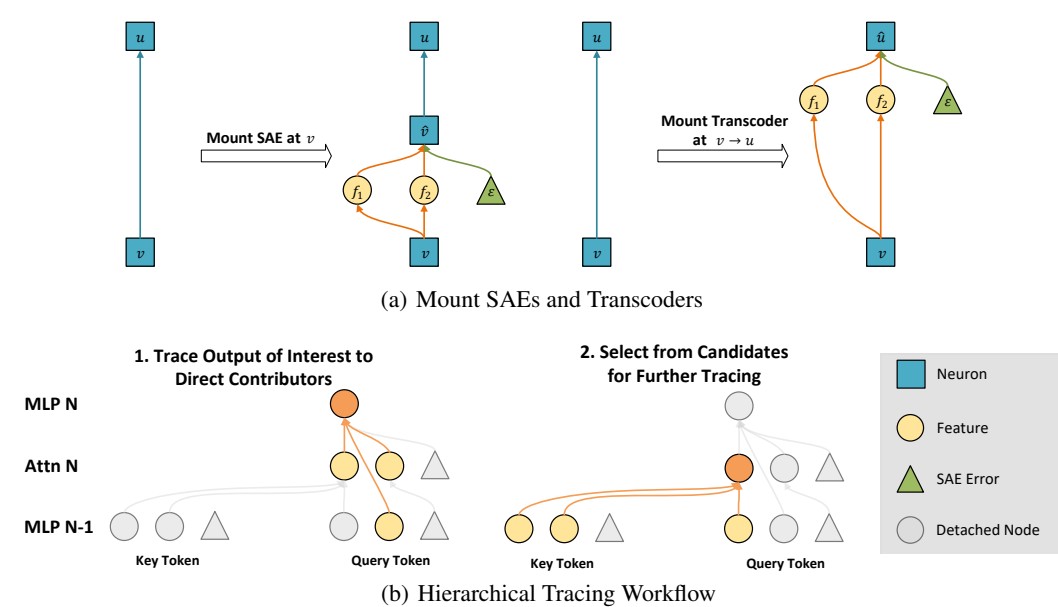

(a) Mount SAEs and Transcoders

(b) Hierarchical Tracing Workflow

Figure 1: (a) Demonstration of mounting SAEs and Transcoders in a computational graph. We insert feature nodes to reconstruct the output, and create SAE error nodes to fix the difference between original outputs and the reconstructions. (b) Our Hierarchical Tracing approach, where we iteratively trace interested output to direct contributors by computing direct effects defined in Equation 5 of all previous features, and select critical candidates for further tracing.

For an SAE $\mathcal{S}$, with encoding function $g_E(x) = \mathrm{ReLU}(W_E x + b_E)$ and decoding function $g_D(x) = W_D x$, we mount the SAE at a specific node $v$ (Figure 1(a)), corresponding to where the SAE was originally trained. This is achieved by:

1. Attaching a feature node $f$ to $v$ via an edge $v \to f$ with the functional dependence $g_E$.

2. Attaching a reconstructed node $\hat{v}$ to $f$ through the edge $f \to \hat{v}$, with the functional dependence $g_D$.

3. Connecting $\hat{v}$ to the original successors of $v$ in the computational graph.

In practice, to account for the imperfect reconstruction ability of SAEs, we create an *SAE error node* as a leaf node (Marks et al., 2024), capturing the difference between $a_v$ and $g_D(g_E(a_v))$. This error term ensures that the forward pass remains consistent with the original computation, while the gradient computation now incorporates the effect of the SAE.

For Transcoders, the process is similar. The Transcoder is mounted at the pre-MLP activation node $v$, and the reconstructed node $\hat{v}$ is connected to the successors of the MLP output node, effectively replacing the original MLP computation with the Transcoder's functionality.

To separate the contributions of different features, we can split the feature node $f$ into multiple nodes, each corresponding to an individual feature extracted by the SAE or Transcoder, allowing for more fine-grained control and interpretation.

**Attributing Nodes to Upstream Candidates.** Once SAEs and Transcoders are integrated into the computational graph, it becomes possible to identify the key upstream nodes that contribute directly to the target output. Previous approaches, such as circuit analysis using activation patching (Wang et al., 2023; Conmy et al., 2023) and attribution patching (Kramár et al., 2024; Marks et al., 2024), have primarily focused on understanding the **indirect effect**—which captures the aggregate influence of intermediate nodes across all possible paths. While these methods are effective at discovering important nodes, they do not guarantee the formation of a coherent and connected subgraph, nor do

they offer a self-contained, interpretable circuit. Additionally, these indirect effects can vary across different tasks due to the complexity and nonlinearity of the underlying neural network functions.

To mitigate such issue, our method centers on computing the **direct effect** of individual nodes by analyzing the path-based gradients (as defined in Equation 4) (Figure 1(b)). This method provides a more precise and interpretable view by isolating the direct contributions of upstream nodes. We define a set of intermediate nodes $\mathcal{V}^I$ as gradient barriers, which block path-based gradients from propagating through these nodes, except for those originating directly from them. The direct effect of a node $v$ on an output node $u$, considering the intermediate nodes $\mathcal{V}^I$, is represented by an attribution score:

$$\text{attr}_v \bigg|_{\mathcal{V}^I} = a_v \sum_{P \cap \mathcal{V}^I = \varnothing} \nabla_v a_u \bigg|_P . \tag{5}$$

Nodes with high direct attribution scores can be identified as critical upstream candidates, providing a more interpretable and connected subgraph for further analysis. In practice, we treat the outputs of the attention heads from SAEs and the features generated by Transcoders as the set of intermediate nodes $\mathcal{V}^I$. Given that the direct effect computation in this setting is relatively straightforward (linear for Transcoders and bilinear-softmax-linear for attention mechanisms), we expect these inter-layer effects to persist across different inputs, enabling a more generalized and robust interpretation of the results.

**Selecting Critical Candidates for Further Tracing.** Once key upstream candidates are identified, the next step is to prioritize the most critical nodes for detailed tracing. This selection is based on their direct attribution scores and their contextual importance within the network. To determine which nodes warrant further analysis, we can employ either of these two strategies:

- Apply thresholds on the attribution scores or use sparsity-promoting techniques to limit the focus to a small subset of paths and nodes (Section 3.3).
- Conduct a more in-depth inspection of the candidates by utilizing top activations of features and direct logit attributions (DLAs), selecting those with the strongest contextual relationships. This selection can be performed either automatically using large language models (LLMs) (Section 4) or manually by human experts (Section 5).

By focusing on the most critical nodes, we reduce complexity while simultaneously enhancing the interpretability of the resulting model, yielding clearer insights into how key features influence the final predictions.

## 3.3 EVALUATING THE GLOBAL NECESSITY OF TRACED RESULTS

After tracing key nodes and subgraphs using Hierarchical Tracing, it is important to evaluate the significance of the traced results from a broader perspective. Specifically, we assess the **necessity** of the traced results by ablation testing. We hypothesize that the removal of key nodes from the traced subgraph should result in a significant drop in model performance if the traced nodes are truly critical to the final output.

For instance, in a text input scenario, we first run Hierarchical Tracing with a sparsity-promoting selector that identifies the top 10 features by its direct effect attribution score from each layer. Next, for a range of values $1 \leq k \leq 40$, we mean-ablate the top-$k$ nodes and measure the probability decrease from the original output. The mean ablation is done by replacing the current activation with the average value at current node across the task. This experiment is compared against a neuronal approach (where intermediate nodes are defined as all model activations that write to the residual stream) and a baseline approach involving the random ablation of activated features. We further mean-ablate a single feature/neuron that is ordered exactly the $k$-th to examine if there are deeper correlations between the effect measured by Hierarchical Tracing and the overall significance.

We evaluate through 500 prompts from the subject-verb agreement task. The results (Figure 2(a)) suggest that ablating the top 20 critical features from the traced subgraph is enough to cause a substantial performance drop, demonstrating that our method successfully isolates vital nodes. Besides,

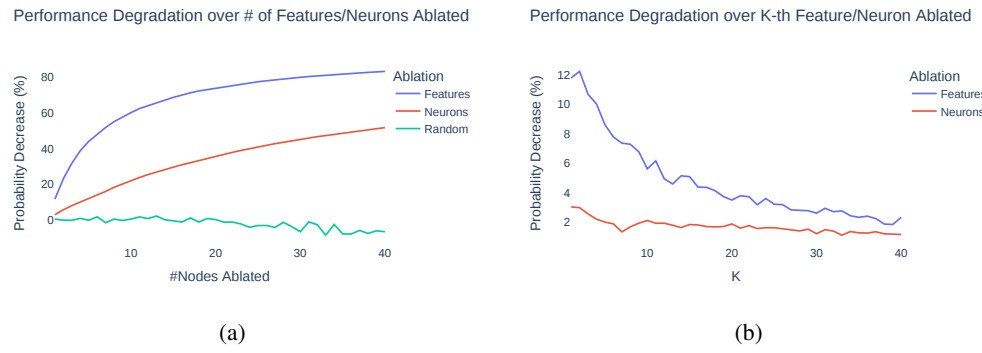

Figure 2: Performance degradation according to the mean percentage probability decrease through prompts of the subject-verb agreement task, when ablating (a) all the top $k$ features or neurons, or randomly-selected $k$ activated features; (b) the exactly $k$-th feature or neuron.

as $k$ increases, Figure 2(b) shows that only ablating the $k$-th node causes a fainter effect, showing the effectiveness of Hierarchical Tracing. Furthermore, our approach consistently outperforms the neuronal approach in identifying key nodes, demonstrating its superior ability to localize critical components in the computational graph.

## 4 FULLY AUTOMATED WORKFLOW FOR INTERPRETABLE CIRCUIT TRACING

To streamline the tracing of interpretable circuits in any model forward pass, we propose a fully automated workflow that combines the hierarchical tracing methodology with GPT-4o, leveraging LLMs to automate the analysis of intermediate activations, select critical nodes, and generate comprehensive explanations of the model's internal mechanisms.

The workflow consists of three main steps:

1. **Feature Interpretation:** We utilize GPT-4o to interpret individual features extracted by SAEs and Transcoders (Bills et al., 2023a; Bricken et al., 2023). By providing GPT-4o with activation contexts, direct logit attributions (nostalgebraist, 2020), and task descriptions, it generates concise explanations of the conditions under which each feature activates, aiding in understanding the semantic or syntactic roles of features.

2. **Candidate Selection:** GPT-4o selects important intermediate nodes that significantly contribute to the model's inference. It considers the current node to trace, candidate upstream nodes with brief explanations, and relevant task information. Following structured interaction guidelines, GPT-4o iteratively selects nodes to trace further, building a coherent and connected subgraph of critical nodes. In practice, to prevent an overload of distractor candidates, we provide GPT-4o with the top 10 features exhibiting the highest attribution scores, as outlined in Section 3.3, and request further selection.

3. **Circuit Interpretation:** Finally, GPT-4o synthesizes the traced information to generate a comprehensive explanation of the model's internal information flow, detailing the progression from low-level token patterns to high-level semantic understanding. This provides a transparent view of the model's decision-making process.

By integrating LLM-based interpretation at each stage, our automated workflow not only identifies critical components within the model but also generates human-readable explanations of how these components contribute to the model's behavior. This approach significantly reduces the need for manual analysis, enabling scalable and efficient interpretability for complex neural networks. Our detailed prompts used for GPT-4o interactions are listed in Appendix D.

We evaluate our LLM-based interpretation using the following criteria:

Table 1: Ratings for automated interpretation workflow

| Criterion | SVA (Simple) | SVA (RC) | IOI | In-Bracket | Induction |
|---|---|---|---|---|---|
| Interpretability | 7.9 | 6.6 | 5.3 | 8.2 | 7.1 |
| Reasonability | 7.9 | 6.7 | 4.9 | 8.4 | 7.2 |
| Generality | 7.4 | 5.5 | 5.0 | 8.7 | 6.8 |

- **Interpretability:** Assessing how clearly the LLM articulates the feature-based information flow in the model forward pass.

- **Reasonability:** Evaluating whether the explanations provided by the LLM are reasonable based on the candidate nodes.

- **Generality:** Evaluating the consistency and coherence of explanation among different prompts of the same tasks.

To assess these criteria, we run our workflow on prompts from three end-to-end tasks: the simple and across-relative-clause variants of the subject-verb agreement task and the indirect object identification task. Additionally, we include two tasks focused on interpreting the formation of a specific feature.

Following this, we enlist experienced human crowdworkers to evaluate each criterion. They manually inspect the inner thought processes, candidate selections, and circuit interpretations provided by the LLMs, assigning ratings based on their assessments. The results in Table 1 show that our automatic approach succeeded in tracing and interpreting information flow in tasks such as subject-verb agreement and intermediate feature formation. Detailed rubrics and interface are listed in Appendix D.2. However, for more complicated tasks like IOI, it falls short of providing a comprehensive summary of the entire circuit. Upon examining the interaction histories, we discovered that our automatic feature interpretation struggles to capture commonalities when the effective context is lengthy, particularly in the case of induction features. We then move on to dive into the process of circuit tracing.

## 5 IN-DEPTH TRACING OF LOCAL AND GLOBAL CIRCUITS

Despite the success of our fully automated approach in generating circuit explanations, it is not so meticulous about the precise information flow. In this section, we turn to manual tracing through local circuits (from an intermediate feature) and global circuits (from the output logits), investigating how contribution from different upstream features affect downstream, and how OV and QK circuits (Elhage et al., 2021; He et al., 2024) collaborates in inter-layer and inter-token information moving.

### 5.1 HOW TRANSFORMERS IMPLEMENT IN-BRACKET FEATURES

Sparse Autoencoders (SAEs) serve as powerful *unsupervised feature extractors* in the expansive hidden activation space of language models. This capability allows us to explore intermediate activations and local circuit discovery, focusing on subgraphs that activate specific SAE features, rather than solely on end-to-end circuit behavior.

We research *In-Bracket* features in the attention blocks of early layers, specifically targeting tokens within brackets (e.g., *deactivated [activated] deactivated*). These features exhibit heightened activation levels with deeper bracket nesting, mimicking finite state automata behavior (Bricken et al., 2023). Our findings reveal an *In-Bracket* feature L1A.F11421 in the SAE trained on layer 1 attention block outputs, referred to as L1A.

**Open-bracket features promote in-bracket activation.** As illustrated in Figure 3(a), we investigate contributions to the *In-Square-Bracket* feature within a template such as "0 0 [1 1 1 [2] 3] 4," focusing on tokens "1", "2", "3", and "4". Our experiments indicate that the activation is primarily driven by an L0M feature activated by the token " [", accounting for 104.1%, 102.6%, and 314.2% of the *In-Square-Bracket* feature's activation for tokens "1", "2", and "3" respectively. Notably, an average of 83.8% of these contributions arises from attention head 1 of L1A, labeled L1A.H1.

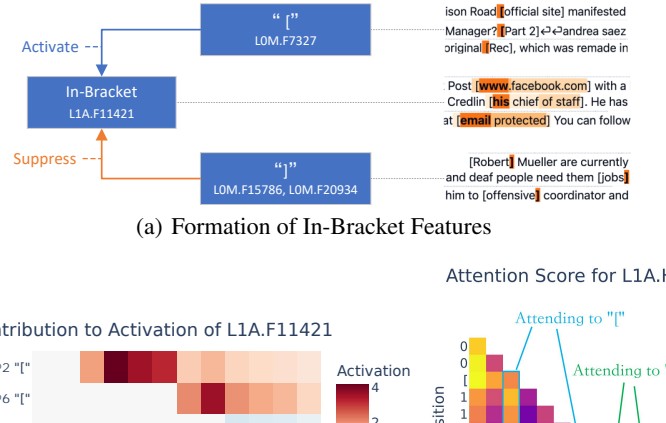

(a) Formation of In-Bracket Features

(b) Contribution to a specific *In-Bracket* feature from each token's bracket features

(c) Attention Score Trends of a Significant *Bracket Head*

Figure 3: (a) *Opening Bracket* features positively contribute to *In-Bracket* features, while *Closing Bracket* features exert a negative influence. (b) The proximity of " [ "s enhances activation of the *In-Bracket* feature. (c) Tokens following " [ "s show strong initial attention to the opening bracket, which diminishes as the sentence progresses, illustrating the trend seen in Figure 3(b).

**Closing-bracket features suppress in-bracket activation.** The *In-Square-Bracket* feature is predominantly inhibited by a "]" feature in L0M, as shown in Figure 3(b), with suppression also mediated through L1A.H1.

**Interpreting QK attention to " [ " and "]".** We analyze the QK circuit of L1A.H1, depicted in Figure 3(c). This head consistently attends to " [ "s and "]"s, irrespective of the current token. This behavior is largely influenced by $b_Q$ in L1A.H1, which engages with the aforementioned bracket features.

In summary, *In-Bracket* features are activated by opening bracket features and inhibited by closing bracket features, resembling the operational dynamics of finite automata. Despite their straightforward functionality, the underlying mechanisms are more complex than initially anticipated.

## 5.2 REVISITING INDIRECT OBJECT IDENTIFICATION CIRCUITS

To explore end-to-end circuits in GPT-2 Small, we investigate the Indirect Object Identification (IOI) task (Wang et al., 2023) using *Hierarchical Tracing*. For example, given the prompt "When Mary and John went to the store, John gave the bag to", GPT-2 predicts " Mary". We refer to this prompt as $s_{\text{Mary}}$, and a variant $s_{\text{John}}$ is created by swapping the names, leading to the same answer. Existing studies often overlook the differences between these templates.

Through the SAE lens, we not only confirm previous findings but also uncover nuanced mechanistic distinctions in their corresponding circuits.

**Feature Circuits Agree with Head-Level Circuits.** Using *Hierarchical Tracing*, we trace the information flow in the IOI task for both $s_{\text{Mary}}$ and $s_{\text{John}}$. We identify critical attention heads within the isolated subgraph and attribute their QK scores to earlier features. Notably, the identified feature circuits show strong correspondence with those derived from attention heads: (1) *Name Mover* features agree with *Name Mover Heads* (L9A.H6, L9A.H9); (2) *Association* features correlate with *S-Inhibition Heads* (L7A.H3, L7A.H6, L8A.H10); (3) *Induction* features match with *Induction*

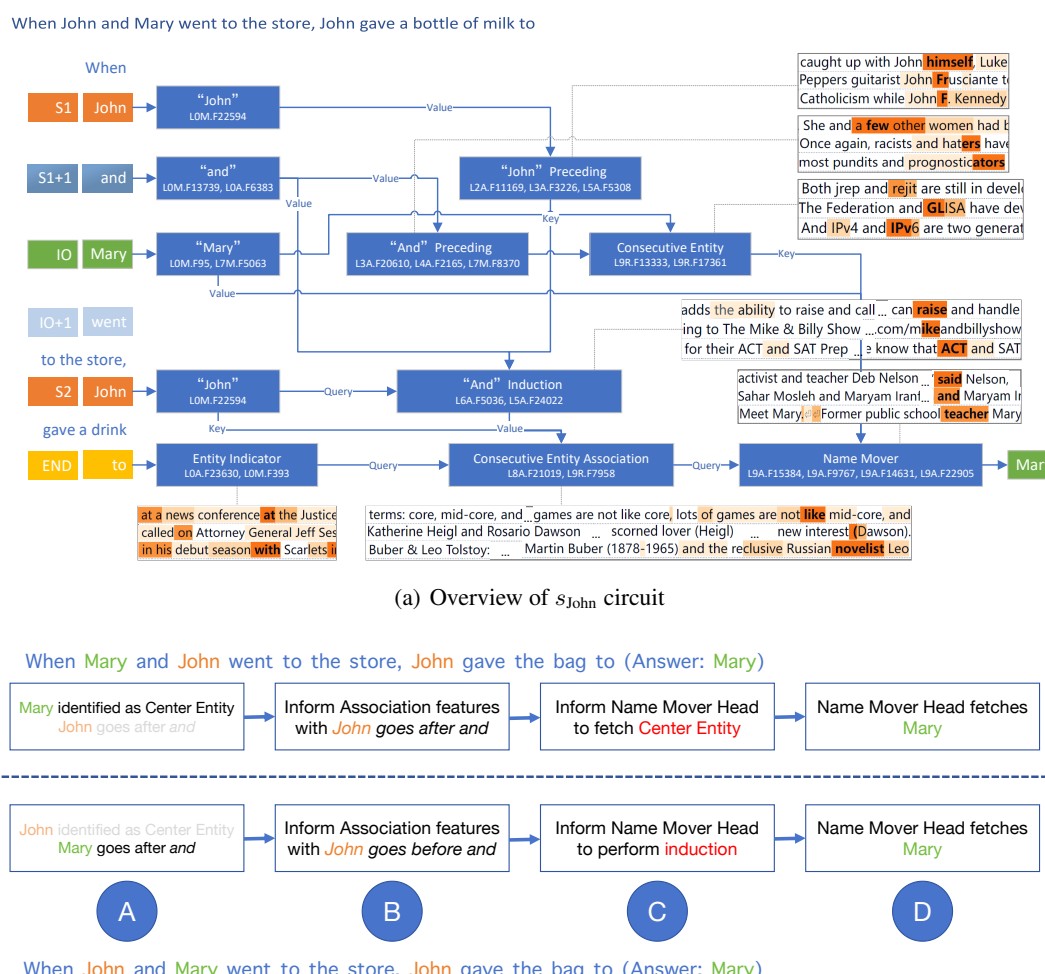

(a) Overview of $s_{\text{John}}$ circuit

(b) Key differences between $s_{\text{John}}$ and $s_{\text{Mary}}$ circuits

Figure 4: In the circuit for $s_{\text{John}}$, the consecutive entity feature (denoted as A in Figure 4(a)) acts as the key vector for Name Mover Heads, allowing them to attend to and replicate the answer entity in the residual stream. This mechanism is ineffective in $s_{\text{Mary}}$, as the correct answer is no longer a consecutive entity (i.e., it follows the token *and*). See Appendix E for detailed interpretations of these examples.

*Heads* (L5A.H5, L6A.H9); and (4) *Preceding* features correspond to *Previous Token Heads* (L2A.H2, L3A.H2, L4A.H1).

**Zooming in SAE Circuits Reveals New Insights.** Our findings indicate that SAE circuits provide richer information than their coarser counterparts, signaling a deeper understanding of language model mechanics. While the attention heads in both $s_{\text{John}}$ and $s_{\text{Mary}}$ show consistency, they operate through distinct SAE features, as illustrated in Figure 4(b).

Focusing on $s_{\text{John}}$, we analyze how GPT-2 predicts " Mary" following the prompt "When John and Mary went to the store, John gave the bag to". The information flow, though simplified, is intricate. We highlight four pivotal feature clusters, as indicated in Figure 4:

A " Mary" is recognized as a Consecutive Entity due to its position following " and".

B The second occurrence of " John" activates an induction feature, enhancing the logit of "and," despite its next token being different.

C The token "to" signifies that the next token is likely an object or entity, activating an association feature to retrieve potential entities that have appeared previously.

D The Name Mover Head receives this information, facilitating the copying of the token " Mary" to the residual stream.

In $s_{\text{Mary}}$, however, the situation diverges significantly. Here, " Mary" first activates a Center Entity feature, which GPT-4 explains as "People or Objects that are likely to be the main topic of the article." The last token aims to associate a previously mentioned entity but is directed to retrieve the Center Entity instead, as the Consecutive Entity Association feature has been inhibited by repeated mentions of " John."

## 6 RELATED WORK

**Mechanistic and Representational Interpretability.** Mechanistic Interpretability (Olah et al., 2020b;a) deems model components, e.g., attention heads and MLP neurons, as *primitives* and explains how they interact with model input and output. This line of research has succeeded in identifying attention-based circuits implementing various NLP tasks (Olsson et al., 2022; Wang et al., 2023; Stefan Heimersheim, 2023). Efforts are also made to interpret polysemantic MLP neurons (Gurnee et al., 2023) and editing information stored in MLP parameters (Meng et al., 2022; Sharma et al., 2024).

By placing intermediate activations at the center of analysis, Representational Interpretability approaches mostly use linear probes to isolate a targeted behavior in a supervised manner (Kim et al., 2018; Geiger et al., 2023; Zou et al., 2023). However, such methods may fail to capture unanticipated behaviors.

**Sparse Autoencoders** stand in between these two approaches. SAEs disentangle features in the model's *hidden activation* (Chen et al., 2017; Subramanian et al., 2018; Zhang et al., 2019; Panigrahi et al., 2019; Yun et al., 2021; Bricken et al., 2023; Cunningham et al., 2023) into more interpretable *primitives* than MLP neurons, in an unsupervised manner. Albeit reconstruction errors, Rajamanoharan et al. (2024); Wright & Sharkey (2024) have proposed to improve SAE training with lower loss and more sparsity.

**Circuit Discovery with SAE Features.** Previous work mechanistically interprets circuits connecting attention heads and MLP neurons (Olsson et al., 2022; Wang et al., 2023; Conmy et al., 2023). As for SAE circuits, He et al. (2024) makes a linear approximation of MLP layers by fixing the gate mask of the non-linear activation function; Marks et al. (2024) estimates the indirect effect of each SAE feature with attribution patching (Kramár et al., 2024), which also makes linear assumption of non-linear functions. In contrast, we refactor our computation graph to be completely linear w.r.t. OV and MLP circuits without approximation.

## 7 CONCLUSION

Our framework employs Sparse Autoencoders (SAEs) to extract fine-grained features from model activations, providing a clearer understanding of how Transformer layers and neurons process information. To address the challenges posed by non-linear MLP structures, we introduce Transcoders, enabling the deterministic tracing of MLP outputs. We further present Hierarchical Tracing, a methodology that allows for both local and global analysis of circuits, facilitating the discovery of how different parts of a Transformer contribute to model behavior.

Through various automatic and manual experiments on tasks like subject-verb agreement and IOI, we have demonstrated the robustness of our approach in isolating critical circuits. The analysis of in-bracket features and indirect object identification circuits showcases the depth of interpretability made possible by using SAEs. Additionally, our automated workflow integrated with GPT-4o streamlines the tracing process, offering scalable and interpretable results.

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

## A  NOTATION SUMMARY

This section summarizes and clarifies the notations used throughout the paper. Each notation is listed with a brief description for reference.

Table 2: List of Notations and Descriptions

| Notation | Description |
|---|---|
| $x$ | Input activation in the model. |
| $\hat{x}$ | Reconstructed input activation. |
| $f$ | Feature vector from SAE or Transcoder. |
| $W_E, b_E$ | SAE or Transcoder encoder weight matrix and bias. |
| $W_D$ | SAE or Transcoder decoder weight matrix. |
| $d_{\text{model}}$ | Dimension of the model's hidden activation space. |
| $d_{\text{SAE}}$ | Dimension of the SAE feature space. |
| $\mathcal{L}$ | Loss function used during training. |
| $\lambda$ | Coefficient for L1 regularization (sparsity loss). |
| $f_i^{\mathcal{S}}$ | Feature $i$ from SAE $\mathcal{S}$. |
| $\mathcal{G} = (\mathcal{V}, \mathcal{E})$ | Computational graph, with vertices $\mathcal{V}$ and edges $\mathcal{E}$. |
| $a_u, a_v$ | Activations at nodes $u$ and $v$ in the computational graph. |
| $P$ | A path in the computational graph. |
| $\nabla_v a_u\big|_P$ | Gradient of activation $a_u$ with respect to $a_v$ along path $P$. |
| $\otimes$ | Aggregation operation over inputs in the computational graph. |
| $\mathcal{V}^I$ | Set of intermediate nodes used as gradient barriers. |
| LXM, LXA | The MLP and attention block of layer X. |
| LXM.FY@Z, LXA.FY@Z | The Y-th feature at token Z from the LXM Transcoder / the LXA SAE. |
| LXA.HY | The Y-th attention head of layer X. |

## B  SPARSE AUTOENCODER TRAINING

Table 3: Statistics of Attention Output SAEs

| SAE | Var. Explained | L0 Loss | Reconstruction CE Score | Reconstruction CE Loss |
|---|---|---|---|---|
| L0A | 92.25% | 29.66 | 99.24% | 3.2327 |
| L1A | 82.48% | 65.57 | 97.19% | 3.2138 |
| L2A | 83.39% | 69.85 | 94.29% | 3.2150 |
| L3A | 69.23% | 53.59 | 87.00% | 3.2173 |
| L4A | 74.91% | 87.35 | 89.99% | 3.2171 |
| L5A | 82.12% | 127.18 | 97.81% | 3.2145 |
| L6A | 76.63% | 100.89 | 94.31% | 3.2158 |
| L7A | 78.51% | 103.30 | 91.32% | 3.2182 |
| L8A | 79.94% | 122.46 | 88.67% | 3.2172 |
| L9A | 81.62% | 107.81 | 89.55% | 3.2187 |
| L10A | 83.75% | 100.44 | 87.70% | 3.2201 |
| L11A | 84.81% | 22.69 | 85.49% | 3.2418 |

We trained Sparse Autoencoders (SAEs) (Section 2.1) on the outputs of all 12 attention layers. For each MLP layer, we trained a Transcoder (Section 2.2), with the residual stream activation before the MLP as input and the MLP output as the label. Below are the training settings:

- **Dictionary size:** Each SAE/Transcoder contains 24,576 features, 32 times the hidden dimension of GPT-2 Small.

- **Optimization:** We use the Adam optimizer with a learning rate of 4e-4 and betas (0, 0.9999) for 1 billion tokens from the OpenWebText (Radford et al., 2019) corpus. Training uses a

Table 4: Statistics of MLP Transcoders

| SAE | Var. Explained | L0 Loss | Reconstruction CE Score | Reconstruction CE Loss |
|-----|----------------|---------|-------------------------|------------------------|
| L0M | 94.16% | 19.59 | 99.65% | 3.1924 |
| L1M | 82.02% | 48.63 | 86.35% | 3.1816 |
| L2M | 86.32% | 50.90 | 81.24% | 3.1851 |
| L3M | 76.55% | 56.91 | 83.43% | 3.1867 |
| L4M | 73.38% | 76.03 | 80.08% | 3.1888 |
| L5M | 73.49% | 84.11 | 84.18% | 3.1881 |
| L6M | 72.79% | 90.34 | 82.85% | 3.1912 |
| L7M | 73.18% | 86.38 | 81.89% | 3.1911 |
| L8M | 74.14% | 87.29 | 83.25% | 3.1913 |
| L9M | 75.89% | 90.08 | 81.89% | 3.1930 |
| L10M | 79.66% | 94.85 | 81.60% | 3.1987 |
| L11M | 80.33% | 79.12 | 77.33% | 3.2169 |

batch size of 4,096 on an NVIDIA A100-80GB GPU, running for 20 hours. Loss functions include reconstruction loss (MSE), sparsity loss (L1 norm of activations, coefficient 8e-5, 1.2e-4 for attention outputs), and ghost gradient loss.

- **Input processing:** Only the first 256 tokens from each sequence are used, discarding sequences shorter than this. Activations are shuffled within an activation buffer.

- **Normalization:** Input activations are normalized to a norm of $\sqrt{768}$ (GPT-2 Small hidden size). The MSE loss is further normalized by the variance of the output along the hidden dimension:

$$\mathcal{L}_{\text{MSE}} = \frac{(x_{\text{normed}} - \hat{x}_{\text{normed}})}{\|\hat{x}_{\text{normed}} - \bar{\hat{x}}_{\text{normed}}\|_2}.$$

- **Weights and biases:** We untie encoder and decoder weights. The decoder bias (pre-encoder bias) is removed to simplify circuit analysis. Decoder norms are constrained to be less than or equal to 1 after each training step.

- **Feature pruning:** Dictionary features are pruned if they have a norm less than 0.99, a maximum activation less than 1, or an activation frequency below 1e-6.

- **Finetuning:** After pruning, we finetuned the decoder and feature activation scaler on the same dataset, with only reconstruction loss applied, to mitigate feature suppression and improve overall reconstruction quality.

## B.1 FEATURE PRUNING

Some SAE features can be overly sparse and activated by very specific tokens, contributing little to overall reconstruction. These trivial features are pruned based on the following criteria:

**Norm less than 0.99:** Useful features tend to have larger norms, as the L1 loss encourages smaller activations. Features without growing norms are pruned.

**Max activation less than 1:** Features with low maximum activation contribute minimally to reconstruction and are often activated in unrelated contexts, making them non-interpretable.

**Activation frequency less than 1e-6:** Features with ultra-low activation frequencies are too local and correspond to specific tokens in very specific contexts. These are pruned if their activation frequency falls below this threshold.

## B.2 FINETUNING TO ADDRESS FEATURE SUPPRESSION

Feature suppression, where loss functions push activation values towards zero, can degrade reconstruction quality. To address this, we finetuned the decoder and feature activation scaler of pruned

SAEs using only reconstruction (MSE) loss, while keeping encoder weights fixed. This finetuning helps restore reconstruction quality and correct any issues caused by pruning.

### B.3 SPARSE AUTOENCODER EVALUATION METRICS

We evaluate the trained SAEs using three metrics:

**L0 Loss:** Average number of features activated per token, measuring the sparsity of the SAE.

**Explained Variance:** Measures the proportion of activation variance accounted for by the SAE:

$$EV = 1 - \frac{\|\hat{y} - y\|_2^2}{\sigma^2(y)}.$$

**Reconstruction CE Score:** The cross-entropy score compares the reconstruction CE loss ($\mathcal{L}_{\text{recons}}$) with the original and ablated CE losses:

$$s = \frac{\mathcal{L}_{\text{recons}} - \mathcal{L}_{\text{ablate}}}{\mathcal{L}_{\text{original}} - \mathcal{L}_{\text{ablate}}}.$$

## C INTERPRETATION TASK DETAILS

In this section, we list details of the language model tasks we mechanistically researched. Table 5 shows the example prompts, answers and outputs of interest in these tasks.

Table 5: Example data from 3 end-to-end tasks and 2 intermediate feature tasks

| Task | Example Prompt | Answer | Interested Output |
|------|----------------|--------|-------------------|
| IOI | "When John and Mary went to the store, Mary gave a bottle of milk to" | " John" | Logit |
| SVA (Simple) | "The girls" | " do" | Logit |
| SVA (RC) | "The friends that the architect likes" | " go" | Logit |
| Induction | "The cuDNN library team is excited to announce ... .We are proud that the cu" | "D" | L5A.F20004 |
| In-bracket | "The Yahoo AP story Man brags he killed Chinese California students [October" | " 17" | L1A.F11421 |

## D DETAILS IN AUTOMATED INTERPRETATION WORKFLOW

This section details the interaction between the direct-effect-based tracer and LLM-based selector. Additionally, we provide information on how crowdworkers rate the interpretability, reasonability, and generality of each sample.

### D.1 TRACER-GPT-4O INTERACTION

For a given forward pass and an interested output, we set the initial target at the interested output, and then iteratively:

- Run the tracer to compute the direct effect of all interested intermediate nodes based on Equation 5;
- Collect the top 10 intermediate nodes, run automatic interpretation, and ask GPT-4o to select one or multiple nodes for subsequent tracing. If multiple nodes are selected, we sum up these nodes and compute a total direct effect for them.

The prompt for automatic interpretation is:

```
━━━━━━━━━━━━ GPT-4o Prompt ━━━━━━━━━━━━
System: You are an expert in Large Language Models and the field of
↪  Mechanistic Interpretability. You're kind to assist in giving
↪  explanation of how language models work.

User: We are analyzing an intermediate activation in a
↪  Transformer-based language model during the forward pass. This
↪  intermediate value may represent neurons in the MLP, the residual
↪  stream, an intermediate layer output, or a specific direction
↪  within these components. The goal is to explain what it signifies
↪  when this value activates (i.e., exceeds 0). You will receive
↪  detailed information about the intermediate value, along with
↪  several contexts where it activates one or more times.

In the contexts, the token where the intermediate value activates will
↪  be denoted as <x, token>, where "x" represents the activation
↪  intensity (1-5, with 5 being the highest), and "token" is the
↪  actual token. Additionally, you will receive the direct logit
↪  attribution of the intermediate value, indicating which tokens it
↪  promotes or suppresses if directly connected to the unembedding
↪  layer. Also, you may receive a task information, which means the
↪  intermediate value is found when the model is performing a specific
↪  task. It does not require this intermediate value to have strong
↪  relevance to the task, but it may help you understand the context
↪  and what we're concerned about better.

Guidelines for generating the explanation:

- Identify shared patterns across contexts where the intermediate value
↪  activates. These patterns could relate to token positioning,
↪  meaning similarity, syntactic roles, surrounding tokens, repetition,
↪  etc.

- Keep in mind that intermediate values from earlier layers often
↪  capture low-level features like syntax or token-level patterns,
↪  while later layers typically reflect higher-level features like
↪  semantics and context. Examine the direct logit attribution for
↪  commonalities in promoted or suppressed tokens, with promoted
↪  tokens more likely to reveal patterns.

- Intermediate values from attention layers often capture token
↪  relationships (e.g., connections with previous tokens or repeated
↪  patterns). Inspect whether similar patterns have appeared earlier,
↪  especially when prior tokens don't trigger activation. Conversely,
↪  intermediate values from MLP layers may focus on individual token
↪  features, though this is not a strict rule.

- Pay special attention to the highest activations (5). Low activations
↪  can be harder to interpret, as they may represent weaker features or
↪  more context-specific behaviors.

Let's begin with the detailed information on the intermediate value,
↪  the activation contexts, and the direct logit attribution.

[DETAILED INFORMATION]
The intermediate value to explain is {node_type} from {position} of
↪  Layer {layer} in a GPT-2 model, which has {total_layer} layers.

[TASK DESCRIPTION]
{task_info}

[CONTEXTS]
Here are the contexts where the intermediate value activates ( denotes
↪  a new line token). Contexts are clipped around the maximum
↪  activation point:
```

```
{contexts}

[DIRECT LOGIT ATTRIBUTION]
The direct logit attribution of the intermediate value is below,
↪  showing the tokens it promotes or suppresses:

Promoted tokens:
{promoted_tokens}

Suppressed tokens:
{suppressed_tokens}

Please respond in the following format:

[THOUGHTS]
Your reasoning process.

[EXPLANATION]
Your concise explanation (maximum 30 words) of the conditions under
↪  which the intermediate value activates, focusing on shared patterns
↪  across contexts.
```

And the prompt for asking GPT-4o to select from candidates is:

```
──────────────── GPT-4o Prompt ────────────────
System: You are an expert in Large Language Models and the field of
↪  Mechanistic Interpretability. You're kind to assist in giving
↪  explanation of how language models work.

User: We are investigating how information flows through a
↪  Transformer-based language model during token generation. Our
↪  process involves tracing output logits back through intermediate
↪  nodes to understand which nodes contribute most to the model's
↪  inference.

In each step, **you** will:
1. Select important intermediate nodes that you believe contribute to
↪  the model's inference.
2. **We** will trace those nodes back upstream to identify vital
↪  upstream nodes.

When multiple nodes are selected in one round, we will trace back based
↪  on the sum of their influence. Only do so if these nodes appear to
↪  have very similar effect. Once you believe enough nodes have been
↪  traced to fully understand the information flow, provide an overall
↪  explanation of how the model generates the next token.

### Interaction Flow:
- We will provide the task description, input prompt, and the next
↪  token.
- In each round, we will trace the current node and provide a list of
↪  candidate upstream nodes. Each candidate will be accompanied by an
↪  explanation in the format: `[ID]: [EXPLANATION]`.
- You can select one or more candidate nodes (separated by commas) that
↪  you think should be traced next by outputting their [ID].
- You can select candidate nodes from previous rounds to trace back if
↪  you believe they are more important or the current tracing branch
↪  is ending.
- If you believe the current tracings are sufficient to explain the
↪  information flow, you can provide an overall explanation by
↪  outputting `[EXPLANATION]`.

### Node Naming Convention:
```

```
1026   The node IDs follow this format: `L{{layer}}{{type_letter}}.{{suffix}}`,
1027   ↪  where:
1028   - `{{layer}}` represents the model layer number (0-11 in GPT-2). Later
1029   ↪  layers capture high-level features (like semantics), while earlier
1030   ↪  layers capture low-level features (like syntax).
1031   - `{{type_letter}}` represents the node type:
1032     - `A`: Attention block.
1033     - `M`: MLP block.
1034     - `R`: Residual stream.
1035   - `{{suffix}}` describes the specific feature, neuron:
       - Example: `F2341@5` refers to feature 2341 at token 5.
1036
1037   ### Additional Node Selecting Guidelines:
1038   - **MLP (M)**: Nodes from MLP blocks capture deeper token-level
1039   ↪  features, often integrating information about syntax and specific
1040   ↪  token patterns. These nodes are essential when the model is
1041   ↪  consolidating information for final token decisions.
1042     - When selecting MLP features, consider if the pattern contributes to
1043     ↪  more complex interactions, such as understanding word roles or
       ↪  generating grammatical forms.
1044   - **Attention (A)**: Attention block nodes capture inter-token
1045   ↪  relationships. Attention nodes often identify key tokens that the
1046   ↪  model focuses on, which can be crucial for understanding
1047   ↪  dependencies.
1048     - When tracing attention nodes, the upstream candidate nodes may
1049     ↪  either contain
1050       - information that is moving to current node (through OV circuit),
1051       ↪  or
1052       - information that determines the attention score (through QK
1053       ↪  circuit), i.e., query and key that determine these two tokens'
       ↪  being attended to each other.
1054     It's worthy to respectively trace back the former and the latter to
1055     ↪  gain a comprehensive understanding of how information flows and
1056     ↪  how the information could flow.
1057   - **Residual Stream (R)**: Residual stream nodes provide a cumulative
1058   ↪  representation of all previous layers' computations. These nodes
       ↪  often contain both low-level and high-level information.
1059     - Trace back residual stream nodes if you want to capture broad
1060     ↪  information about the model's processing across layers.
1061   - **Early Layers**: Early layers (e.g., L0-L3) often capture low-level
1062   ↪  patterns such as token identities or syntactic rules. When you
1063   ↪  trace to early layers, consider returning to later layer nodes
       ↪  (maybe from previous rounds) to gain a more comprehensive
1064   ↪  understanding of the information flow, e.g. going back to a high
1065   ↪  layer attention node and change from OV to QK circuit.
1066
1067   **Important Considerations**:
1068   - Prioritize nodes in higher layers if you are tracing broad semantic
       ↪  patterns, as they integrate more abstract features.
1069   - Trace MLP nodes when you suspect that the model is resolving
1070   ↪  token-level choices, like grammar or token disambiguation.
1071
1072   ### Explanation Guidelines:
1073   When providing an explanation, ensure you construct a clear
1074   ↪  **information flow trajectory** that highlights critical nodes and
1075   ↪  how they contribute to the model's decision-making. Here's what to
1076   ↪  include:
1077   - **Overall Information Flow**: Provide a high-level summary of how
       ↪  information flows from the earlier layers to the final decision,
       ↪  emphasizing how the traced nodes combine to produce the next token.
       ↪  Highlight the progression from low-level to high-level features
       ↪  (e.g., syntax, semantics).
1078
1079
```

```
- **Critical Nodes**: Identify the most significant nodes that
↪  influence the token generation. Explain why these nodes are crucial
↪  in shaping the output and how their roles evolve as the model
↪  processes deeper layers.
- **Inter-node Dependencies**: Describe how the selected nodes interact
↪  with each other. Highlight any relationships between tokens captured
↪  by attention nodes or features consolidated in MLP blocks. Focus on
↪  dependencies such as subject-verb agreement or other
↪  syntactic/semantic patterns.
- **Node Influence**: Assess the strength of each node's influence on
↪  the overall output. For instance, explain whether a residual stream
↪  node has cumulative significance or whether an attention node
↪  reveals a key relationship that drives the next token choice.
- **Conclusion**: Based on the traced nodes, conclude how the model
↪  arrived at its final decision. Summarize the critical steps and
↪  transformations that occurred throughout the layers, noting whether
↪  additional tracing is needed or if the information flow is fully
↪  understood.

### Response Format:
Your responses should follow this format:

[THOUGHTS]
Your brief thought process.

[NODE] / [EXPLANATION]
The selected node ID(s), separated by commas (e.g., `L5A.F123@3,
↪  L7M.N234@6`). Do not append any text including trailing `.` after
↪  the last selected node. / Your explanation of why these nodes are
↪  significant in understanding the mechanism.

You should respond with either [NODE] or [EXPLANATION] in each round,
↪  but not both.

### Task Description:
{task_info}

Input prompt: "{input_prompt}"
Next token: "{next_token}"

### Round 1: (Max {max_rounds} Rounds)

Current Node to Trace:
{target}

Candidate upstream nodes:
{candidates}

Please select the most relevant node(s) to trace and provide their
↪  ID(s). If you believe the current tracings are sufficient to
↪  understand the mechanism, provide an overall explanation of the
↪  information flow.
```

## D.2 HUMAN EVALUATION

We ask human experts to give ratings (1-10) of each result regarding interpretability, reasonability, and generality, based on the task, the explanation given by LLM, and the detailed conversation. Our ratings are based on the rubrics below:

**Interpretability Rubric:**

- **9-10:** Explanations are exceptionally clear and detailed, providing a thorough understanding of the feature-based information flow, and perfectly explaining information from different sub-circuits.
- **7-8:** Explanations are mostly clear, with minor ambiguities that do not significantly hinder understanding.
- **5-6:** Explanations are somewhat clear but lack detail, making it difficult to fully grasp the information flow.
- **3-4:** Explanations are unclear, with significant gaps in information that obscure understanding.
- **1-2:** Explanations are incomprehensible or irrelevant, providing no useful insight into the information flow.

**Reasonability Rubric:**

- **9-10:** All explanations are highly reasonable and well-supported by the candidate nodes, demonstrating strong logical coherence.
- **7-8:** Most explanations are reasonable, with few unsupported claims or logical inconsistencies.
- **5-6:** Some explanations are reasonable, but several claims lack sufficient support or show inconsistencies.
- **3-4:** Explanations are largely unreasonable, with many unsupported claims and significant logical gaps.
- **1-2:** Explanations are completely unreasonable and full of speculations.

**Generality Rubric:**

- **9-10:** Explanations are highly consistent and coherent across different prompts and tasks, demonstrating a robust understanding of the model's behavior.
- **7-8:** Explanations are mostly consistent, with minor variations that do not significantly affect overall coherence.
- **5-6:** Explanations show some consistency, but notable discrepancies exist between different prompts and tasks.
- **3-4:** Explanations are largely inconsistent, with many contradictions between different prompts and tasks.
- **1-2:** Explanations are completely inconsistent and incoherent, lacking any meaningful connection across prompts and tasks.

Figure 5 shows the interface to obtain the ratings in Table. 1.

# E  ADDITIONAL EXPLANATION OF THE IOI CIRCUIT

This section provides a detailed explanation of the feature circuits identified in the $s_{\mathrm{Mary}}$ and $s_{\mathrm{John}}$ examples by elaborating on the functionality of key features in both cases.

Key features in the $s_{\mathrm{John}}$ circuit (Figure 4(a)):

- *"John", "and", and "Mary" Features*: These features simply indicate that the current token corresponds to "John", "and", or "Mary", respectively.
- *Entity Indicator Features*: Activated on prepositions or transitive verbs, these features suggest that the next token is likely an entity.
- *"John" Preceding Features*: These features gather information from the previous token and indicate that the token preceding the current one is "John."
- *"And" Preceding Features*: Similar to the "John" preceding features, these collect information from the previous token and imply that "and" is the token directly preceding the current one.

**Annotating Sample 1**

**Task Info**

| | |
|---|---|
| **Task Name:** | Subject-Verb Agreement (Simple) |
| **Task Description:** | We are investigating how language models decide to use third-person singular verbs in English sentences. |
| **Interested Output:** | the logit of the next token (output) |

**Result for Sample 1**

| | |
|---|---|
| **Input Prompt:** | The girls |
| **Answer:** | do |

**Explanation:**

In this information flow tracing through a Transformer-based language model, we've explored how the model generates the token "do" following "The girls." Initially, nodes such as L10M.F3813@1 and L9M.F3530@1 are activated to recognize the plural noun "girls," signifying a group entity. These nodes function by capturing the semantics of plurality and actions typical to group nouns.

Further downstream, the focus shifts to identifying contexts that specifically require plural verb forms. L8M.F20406@1 contributes by linking plural nouns to auxiliary verb preferences, marking the necessity for plural verbs when encountering subjects like "girls."

As we traced back through L6M, L4M nodes, and further, nodes such as L2M.F18923@1, and L3M.F12943@1 also played a pivotal role. These nodes solidify the understanding of "girls" as a collective identity, emphasizing their role within sentence structures requiring plural verb forms. This consolidation of group entity recognition supports the decision behind the model's output of "do" rather than a singular verb form like "does."

By integrating patterns from the earlier layers, where basic syntactic and semantic roles are identified, and leveraging the deeper, more abstract features from higher layers, the model accurately resolves the need for coherence in the verb choice following "The girls." Through this analysis, it is evident how the model uses a layered approach to move from recognizing entities to making more complex grammatical decisions, illustrating the robust handling of plural subject-verb agreement.

**Detailed Chat History**

☑ Hide Guidelines

**System:**

You are an expert in Large Language Models and the field of Mechanistic Interpretability. You're kind to assist in giving explanation of how language models work.

**User:**

**Round 1: (Max 20 Rounds)**

Current Node to Trace: output: the logit of the next token

Candidate upstream nodes:

1. L11M.F8742@1 : The intermediate value activates when identifying or categorizing entities or subjects within descriptive or narrative contexts. Currently activated at token 1: " girls"
2. L10M.F3813@1 : Activates for plural noun subjects representing entities engaged in significant actions or roles, especially in descriptive or narrative contexts. Currently activated at token 1: " girls"
3. L10M.F2495@1 : Activates when tokens represent primary actors or subjects, especially those implying actions, agency, or defining roles within a sentence. Currently activated at token 1: " girls"
4. L11M.F3535@1 : Activates in contexts with structured date-time formats emphasizing timestamps, recognizing their textual coherence role, supported by function and punctuation token promotion. Currently activated at token 1: " girls"
5. L10M.F20221@1 : Activates when tokens signify pivotal or ongoing events requiring emphasis within thematic structures, often marked by contrast, listing, or detail-focused segments. Currently activated at token 1: " girls"

Figure 5: The Interface for Annotating Circuit Interpretation

- *Consecutive Entity Features*: These features combine the "Mary" features with "And" Preceding features, suggesting that the current token follows an [A] and [B] pattern, where both [A] and [B] are entities.
- *"And" Induction Features*: These features attend to the token "and" by matching sequences *S1* and *S2*, implying that "and" follows "John" in the sentence structure.
- *Consecutive Entity Association Features*: Utilizing structural information from the "And" Induction features, these features identify the entity following "and" by attending to the Consecutive Entity features in the Name Mover heads.
- *Name Mover Features*: These features complete the final step by transferring the information associated with "Mary" from the targeted Consecutive Entity token.

Key features in the $s_{\text{Mary}}$ circuit (Figure 6):

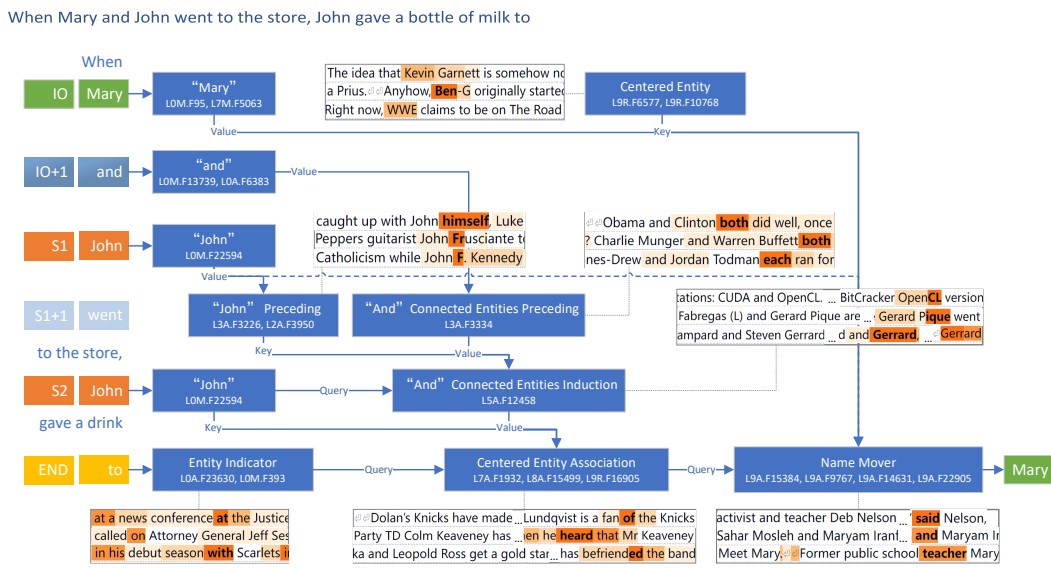

Figure 6: Overview of the $s_{\text{Mary}}$ circuit.

- *"John", "and", "Mary", Entity Indicator, and "John" Preceding Features*: These features behave similarly to their roles in the $s_{\text{John}}$ circuit, marking tokens and their relationships.

- *Centered Entity Features*: These features are activated on the first appearance of a significant name or object, flagging it for potential future reference.

- *"And"-Connected Entities Preceding Features*: These features collect information from several previous tokens (primarily "and"), indicating an [A] and [B] entity pattern before the current token.

- *"And"-Connected Entities Induction Features*: These gather information from the "And"-Connected Entities Preceding features by again matching sequences *S1* and *S2*.

- *Centered Entity Association Features*: Leveraging the structural information from the "And"-Connected Entities Induction features, these features identify the entity preceding "and" by attending to the Centered Entity features in the Name Mover heads. Unlike Consecutive Entity features, Centered Entity features do not account for the "and" token that follows. However, this behavior is still reasonable, as a previous Centered Entity could also serve as a valid answer if present before the indirect object.

- *Name Mover Features*: As in the $s_{\text{John}}$ circuit, these features perform the final step of transferring the information about "Mary" from the targeted Consecutive Entity token.

