# OpenReview forum: "Automatically Identifying and Interpreting Sparse Circuits with Hierarchical Tracing"
_ICLR.cc/2025/Conference — ICLR 2025 Conference Withdrawn Submission_

### Official Review · Reviewer_3qVX · 2024-10-17

**Soundness:** 2
**Presentation:** 3
**Contribution:** 4
**Rating:** 5
**Confidence:** 3

**Summary:**

This paper presents a process to construct interpretable computational graphs of SAE features that could help understand the algorithms implemented by Transformers.

The main method it introduces is Hierarchical tracing, which approximates Transformers with a succession of Transcoders, and then uses gradients in this approximated graph to select the most relevant nodes and edges between nodes.

These graphs can then be further pruned and annotated, either by hand or using GPT-4o.

This process is applied to generate explanations for simple behaviors of GPT-2-small.

For simple very behaviors resulting in very small graphs, the graphs are relatively faithful (in that ablating its nodes results in large performance degradations) and GPT-4o can give plausible annotations to the nodes.

For slightly more complex behaviors and larger graphs, GPT-4o struggles to find plausible pruning and annotate, but manual annotation and pruning results in plausible graphs (the faithfulness of such graphs remains to be determined).

**Strengths:**

* Using Transcoders to build graphs to explain behaviors seems promising.
* The idea of using gradient-based is interesting and this approach to graph pruning looks efficient and scalable.
* The methods and results are clearly presented. The Figures are clean and helpful.
* The explanation for the indirect object identification circuit is probably the most complex explanation of a behavior observed in a Language model.
* The process presented could be extended to be more scalable, and since the graphs are supposed to be causal explanations, they could in principle be evaluated using path ablation methods.
* The process is applied to explain multiple behaviors in non-toy Transformers.
* There are some experiments evaluating the faithfulness of the some of the generated explanations.

**Weaknesses:**

**Lack of materials**

* Not enough material is provided to enable a reproduction of the results. No code is provided, and no hyperparameters are provided for Hierarchical tracing. No explanation of how "manual tracing" was conducted is described.
* Only some results are presented for each behavior studied. Only for subject-verb agreement is any faithfulness measurement provided. Only for in-bracket activation is percentage of feature's activation provided. Only for in-bracket activation and indirect object identification are graphs provided. More results for each methods would allow readers to understand how faithful explanations are for more complex behavior, and it would also allow readers to understand what the graphs look like for the only behavior for which faithfulness was measured.
* Examples of the graphs produced by GPT-4o would also allow the reader to get a better sense of how good these are (the human ratings are not as informative as examples of graphs).

**Lack of evidence for faithfulness or usefulness**

* The IOI graphs are provided as is, without any evaluation of their faithfulness, which is especially concerning since they were built using manual tracing.
* Previous work has shown that the percentage of loss explained by graph explanations (as measured with causal scrubbing) was often low. Applying similar metrics to the explanations provided here might highlight a lack of faithfulness of the explanations. Because this paper does not present such measurements, nor any downstream activations, future work is required to determine if the process describe by the paper produced explanations that have other qualities than their plausibility.

**Unjustified implications of high explanation quality**

The paper implies that the annotations and explanations are much more reliable than they are shown to be. The paper only provides weak evidence of faithfulness, only providing strong evidence that the method is helpful to produce highly plausible explanations. This is not enough to claim a superiority in explanation quality or the ability to actually understand how a behavior was implemented, especially given that previous works have shown that interpretability illusions are common when plausibility is used as the main criteria to evaluate explanation quality. Here is a non-exhaustive list of places where this seems particularly problematic:

* "This provides a transparent view of the model’s decision-making process." (line 317) is misleading given the potentially high level of unfaithfulness of the explanations.
* The last sentence of the abstract might need to be revisited to not imply a level of explanation faithfulness that is higher than demonstrated, especially since some prior work actually provided a better justification of the faithfulness of their methods.
* The "in-bracket" SAE feature is unlikely to be fully explained by "is this token in brackets", as the activations vary considerably between the different tokens in brackets. Adding a sentence in section 5.1 explaining this approximate nature would help.

**Minor comments**

* The subject-verb agreement tasks are not described.
* The distribution used for the in-bracket task is not described.
* $y$ in equation 2 is undefined.

**Overall assessment**

This paper would be above the bar if the faithfulness limitations were highlighted and if either easily usable code was released, or if the lacking materials was released, or if the faithfulness measurements was improved.

**Questions:**

* Would it be possible to get the missing materials, and will code be released? (see weaknesses)
* What is the definition of the metric described on line 376 (proportion of feature activation)?
* What is the exact process used to generate the IOI graph? (My current understanding of the paper is that I have to trust that the authors did not simply pick some relevant SAE features and draw plausible edges between them.)

---

### Official Review · Reviewer_ETsY · 2024-10-20

**Soundness:** 2
**Presentation:** 2
**Contribution:** 2
**Rating:** 3
**Confidence:** 3

**Summary:**

The paper presents an automated framework for interpreting Transformer models. By combining Sparse Autoencoders and Transcoders, the author is able to extract fine-grained features and decompose MLP outputs for enhanced circuit analysis. It introduces a hierarchical tracing methodology to isolate key feature subgraphs, supported by an automated GPT-4o-based workflow for scalable analysis. Experiments confirm its effectiveness in isolating critical circuits and assessing their significance through ablation testing. However manual tracing remains essential for detailed analysis of specific circuits like in-bracket features and indirect object identification tasks.

**Strengths:**

1. The use of SAEs and Transcoders to address the interpretability challenges, offering a more fine-grained approach to circuit analysis.
2. The introduction of a scalable tracing approach for identifying interpretable circuits provides deeper insights into model internals.

**Weaknesses:**

1. The framework's effectiveness relies heavily on the interpretability of features extracted by SAEs and Transcoders.
2. Manual tracing remains necessary for in-depth analysis, limiting scalability for large models or complex tasks.
3. The framework faces challenges in providing comprehensive summaries for more complex tasks.
4. There is a lack of theoretical analysis or insights into the effectiveness of the proposed methods.

**Questions:**

1. What objective criteria are used to evaluate the interpretability of extracted features, and how is their interpretability assessed?
2. How does the performance of Transcoders compare to other non-linearity handling methods, and are there experiments validating their use in other non-linear components?
3. Can the tracing process be further automated, and how does the need for manual tracing change with task complexity?

---

### Official Review · Reviewer_gq1s · 2024-10-23

**Soundness:** 2
**Presentation:** 2
**Contribution:** 3
**Rating:** 1
**Confidence:** 5

**Summary:**

This paper presents a new method for analyzing Transformer circuits through the use of Sparse Autoencoders (SAEs) and Transcoders. It details a technique for extracting features from model activations and addressing MLP output non-linearities via deterministic circuit tracing. A Hierarchical Tracing methodology is introduced, aimed at isolating interpretable circuits on both local and global scales, which provides insights into tasks like subject-verb agreement and indirect object identification. Additionally, the paper integrates a fully automated workflow that utilizes GPT-4o, intended to enhance the scalability of circuit analysis.

**Strengths:**

The paper is well-written, enhancing its readability and clarity.

The visualizations are clear and insightful, making the complex concepts easy to understand.

**Weaknesses:**

1. I cannot find any comparisons with baseline methods in the paper. Such comparisons are essential for objectively assessing the effectiveness and advantages of the proposed approach.

2.  The paper could benefit from more comprehensive ablation studies, particularly regarding the hyperparameters like $\lambda$ in Equations 1 and 2.

3. The experimental section of the paper is limited to demonstrations with a few examples rather than extensive benchmarks across diverse datasets. Expanding the experiments to include large-scale benchmarks would provide a more thorough validation of the method's effectiveness and generalizability.

4. The method is only based on the GPT-2 model, without consideration for other widely used or state-of-the-art models like LLaMA or Mixture of Experts. This limitation narrows the significance of the paper, as it does not demonstrate whether the approach can be effectively applied to newer or more complex architectures that are currently prevalent in the field.

**Questions:**

See the Weakness

---

### Official Review · Reviewer_EBrR · 2024-11-01

**Soundness:** 2
**Presentation:** 3
**Contribution:** 2
**Rating:** 5
**Confidence:** 5

**Summary:**

The work introduces a novel approach for sparse circuit identification within LLMs. The proposed method is based on applying SAEs and Transcoders to the model and identifying task-important features within them. Identification is based on the direct effect of features, which are then filtered using both a threshold-based and a LLM approach. Findings mainly confirm previous knowledge found in the literature.

**Strengths:**

- Methodology and formulation are clearly introduced, and experiments are carried out rigorously. However, I would have preferred to see how this method performs against existing ones.
- The work presents a novel approach to sparse features circuit identification through SAEs and Transcoders.
- The work adopts both a simple threshold-based approach and a novel, more complex LLM approach for finding relevant features
- The authors present two case studies on two language tasks identifying circuits that implement them in LLMs

**Weaknesses:**

- Issues regarding indirect effects are not justified (215 -217). To propose a new method based on direct effects, I'd like to understand why it's necessary and what the limitations of indirect effects methods are.
- There's no comparison with indirect effects based methods. I'd like to answer the question, "Which method is best?"
- Evaluation is carried out only considering the necessity of the identified important features. Why is sufficiency not considered in this case? Using already adopted metrics such as faithfulness and completeness would have been more useful in this case, or at least provide a reason why they were not considered.
- There's no comparison between threshold and LLM approaches to find important features. I'd like to see how they perform in terms of sufficiency and necessity.

**Questions:**

1. Can you clarify the rationale for focusing solely on direct effects?
2. Would you consider a comparative analysis with indirect effect-based methods?
3. Why does the evaluation focus only on the necessity of features and not their sufficiency?
4. Could you clarify the lack of comparison between threshold and LLM-based methods?

---

### Official Review · Reviewer_vtyB · 2024-11-05

**Soundness:** 2
**Presentation:** 3
**Contribution:** 2
**Rating:** 3
**Confidence:** 3

**Summary:**

This paper presents a framework for Transformer circuit analysis, leveraging Sparse Autoencoders (SAEs) and Transcoders. The authors develop SAEs to capture fine-grained features from model activations, while Transcoders enable deterministic tracing through non-linear MLP layers. The proposed Hierarchical Tracing methodology isolates and interprets circuits at both local and global levels, allowing insights into tasks such as subject-verb agreement and indirect object identification. Additionally, an automated workflow incorporating GPT-4o is introduced to scale circuit analysis. The experimental results demonstrate that this approach effectively uncovers Transformer model behaviors by tracing individual SAE-derived features. This framework offers improved interpretability of model mechanics and shows robust performance across various tasks. Results reveal insights into activation flows within Transformer layers, providing an understanding of the model's response to linguistic structures.

**Strengths:**

1. The paper presents an interesting framework that combines Sparse Autoencoders and Transcoders to analyze Transformer circuits, proposing a new hierarchical approach for understanding model behavior.
2. The interpretability of large language models (LLMs) is an important and timely problem.

**Weaknesses:**

1. **Transferability to Other Models**: The authors mentioned that the Sparse Autoencoders (SAEs) are trained specifically on GPT-2 to decompose its residual stream modules, including Word Embedding, attention, and MLP outputs. Can the framework transfer to other Transformer models with different architectures?
2. **Novelty and Scope**: The use of Sparse Autoencoders (SAEs) for feature extraction is well-explored in the interpretability domain, and prior works have leveraged SAEs for fine-grained feature decomposition in neural models. Could the authors clarify what novel insights their application of SAEs brings to Transformer circuit analysis beyond existing approaches? Specifically, how does this method offer interpretability that surpasses traditional linear probes or other SAE-based frameworks?
3. **Quantitative Evaluation**: Although the paper includes experiments, the results are primarily qualitative. Additional quantitative analysis comparing interpretability and accuracy trade-offs with similar approaches (e.g., linear probes or standard SAE circuits or model editing methods [1, 2]) would make the findings more robust.
4. **Automated Workflow Limitations**: While the automated workflow using GPT-4o is a strong addition, its effectiveness is not fully substantiated. The authors should provide clearer benchmarks or comparison metrics to illustrate how it scales relative to other interpretability frameworks.

**Questions:**

Please refer to the weaknesses.

---

### Note · Authors · 2024-11-23

**Comment:**

I have read and agree with the venue's withdrawal policy on behalf of myself and my co-authors. We thank all the reviewers for their time and constructive feedback.

**Withdrawal Confirmation:**

I have read and agree with the venue's withdrawal policy on behalf of myself and my co-authors.